# Characterization of Great Toe Extension Strength Using ToeScale—A Novel Portable Device

**DOI:** 10.3390/s24154841

**Published:** 2024-07-25

**Authors:** Raghuveer Chandrashekhar, Luciana Fonseca Perez, Hongwu Wang

**Affiliations:** 1Department of Occupational Therapy, College of Public Health and Health Professions, University of Florida, Gainesville, FL 32611, USA; rchandrashekhar@phhp.ufl.edu; 2John Crayton Pruitt Family Department of Biomedical Engineering, University of Florida, Gainesville, FL 32611, USA; lu.fonsecaperez@ufl.edu

**Keywords:** great toe extension strength, force development curve, grip strength, machine learning

## Abstract

Great toe strength (GTS) weakness is linked to declines in balance and mobility. Accurately assessing GTS, particularly great toe extension strength (GTES), is often neglected in clinical evaluations due to cumbersome and subjective methods. This study aims to characterize the force development curve output from the ToeScale and examine GTES variations with age, sex, BMI, and grip strength (GS) using traditional analyses and machine learning (ML). We conducted a pilot, cross-sectional feasibility study with convenience samples. We assessed GS using a hand-grip dynamometer and GTES using the ToeScale. The data analysis included descriptive statistics, correlations, independent samples *t*-tests, and accuracy and area under the curve (AUC) scores for three ML models. Thirty-one participants (males: 9; females: 22), 14 young (18–24 years) and 17 older (>65 years) adults, participated in the study. Males had significantly higher peak GTES than females in both age groups. The associations of GTES parameters with BMI and GS varied by age and sex. The ML model accuracies and AUC scores were low–moderate but aligned with traditional analyses. Future studies with larger samples and optimized ML models are needed.

## 1. Introduction

The strength and range of motion of the great toe/first metatarsophalangeal joint (1st-MTJ) are critical for normal walking and balance [1,2,3,4]. The great toe flexors and extensors play a significant role in sensation/proprioception, generating propulsive forces, weight bearing, foot clearance, and supporting the foot arch during different phases of gait and other functional activities [5,6,7,8,9]. Reduced muscle strength, limited range of motion, and/or loss of sensation in the great toe can affect an individual’s balance and their ability to walk and perform activities of daily living (ADLs), and reduce participation in community activities, resulting in a reduction in the individual’s quality of life (QoL) [10,11,12,13,14,15,16,17]. Furthermore, various pathological conditions (neurological and non-neurological), such as peripheral neuropathy, radiculopathy, Charcot–Marie–Tooth disease, and hallux/toe deformities, have also been associated with the atrophy and/or reduction in the strength of great toe muscles [18,19,20,21,22,23,24]. Multiple studies have also reported an association of great toe strength (GTS) with age and sex, making it a potential clinical biomarker that could be used to detect or evaluate the onset and progression of different health conditions [7,25,26,27,28,29].

Despite its critical biomechanical and functional roles, GTS, particularly great toe extension strength (GTES), is often overlooked during routine clinical practice due to the lack of a reliable and robust tool for GTES measurement and the subjective nature of existing methods [30,31,32,33]. To address the limitations of existing GTS measurement methods and devices and the clinical need for their improvement, we recently developed the ToeScale, a novel, portable device [34,35]. A preliminary study using the ToeScale and comparing it with the results from a manual muscle test (MMT) indicates an association between the values obtained by both methods, and the ToeScale shows stronger discriminative ability than the MMT [35]. In addition, the ToeScale output has a force development curve over time, showing not just a peak force that can be measured using a hand-held dynamometer, but also information on how the force was developed over time. However, the characterization of this ToeScale output curve for clinical use is yet to be further examined.

Thus, the primary aim of this study is to characterize the output force development curve of the ToeScale by manually hand-picking features from the force development curve based on its relevance to the muscle’s physiological performance and assess how the different GTES parameters related to demographic variables such as age and sex and grip strength. The secondary objective is to explore the feasibility of employing machine learning (ML) methods to characterize ToeScale output curves by age and sex based on the time series data of the force development curve.

## 2. Materials and Methods

### 2.1. Study Design, Inclusion Criteria, and Measurement Protocol

In this cross-sectional study, data were collected from a convenience sample of younger (aged between 18 and 24 years) and older adults (>60 years of age). As this study focuses on the preliminary characterization and analysis of GTES as well as the feasibility of using the ToeScale in a community setting, we did not have any exclusion criteria for the participants. All participants completed a demographic and physical activity questionnaire and a grip strength assessment using a Jamar handgrip dynamometer (Performance Health Supply Inc., Warrenville, IL, USA) [36]. Finally, the great toe extension strength (GTES) was assessed using the ToeScale [34,35], a novel, portable device. GTES was measured with the participants seated with their knee and ankle at 90° as shown in Figure 1 below. In this seated position, the participants were instructed to raise their great toe, i.e., extend it against the toe cap of the ToeScale, as hard as possible and try to aim for a higher force for 10 s continuously. We selected a 10-s duration for the trials based on our previous work [35]. The measurement of GTES using the ToeScale is shown in Figure 1.

### 2.2. Great Toe Extension Strength Characterization and Classification

The device recorded the force in kilograms with a sampling frequency of 50 Hz, resulting in a force–time curve with 500 data points for each participant, with one column representing time and the second column representing GTES in Kg, and the data were saved as a text file. For the first objective, the force–time curve was characterized using MATLAB. The different parameters/features of the GTES curve extracted included peak force; time to 80% of the peak, i.e., rise time; average force after reaching the 80% of the peak force; percentage of data points in each trial equal to or above the average value; and the RFD (80% peak force divided by the rise time). The different GTES parameters extracted from the GTES curve are shown in Figure 1 below. These parameters were then compared across age groups/sexes. This study chose 80% of the peak force as the cut-off to calculate rise time. The literature reports that all muscle/muscle group motor units are recruited at 80% of the maximum voluntary contraction, emphasizing its significance [37]. Using 80% of the peak force as the cut-off would also help standardize the rise time calculation. The other clinically meaningful measure of GTES is RFD, which has been associated with postural stability and could be a potential biomarker for acute muscle damage and exercise-induced fatigue [38]. RFD in this study is defined as the rate of rise in toe strength per unit time (Ns-1) to reach 80% of the peak toe strength.

For the second objective, the Python 3.0 [39] programming language was used to apply machine learning (ML) algorithms to classify the GTES force–time curves based on age and sex. The units of all force data, i.e., GS and GTES force–time curves, were converted to newtons before the analyses. As this is a preliminary analysis, only supervised ML models [40,41,42] were applied to the GTES force–time curves using age and sex as the two target variables for classification. We applied k-nearest neighbors (k-NN), support vector machine (SVM), and random forest (RF), as these methods are frequently used models due to their unique advantages for smaller time-series datasets [41]. The k-NN classifier is advantageous with small datasets due to its simplicity and effectiveness with limited data [43,44,45], the RF classifier is more robust with handling overfitting and effective in handling feature importance [46], and lastly, the SVM classifier is also robust with handling overfitting and non-linear relationships [42,47]. Before running any of the ML models, the GTES data with missing data were imputed based on the time-point within the 10 s trial where the data were missing. The only missing data observed in the GTES trials were those which had <10 missing data points at the end of the 10 s trial, which were imputed with the last available value to ensure that all datasets had an equal number of data points. Once the data were cleaned and imputed, the ML models were applied directly to the raw dataset (500 data points), without using the manually extracted features in the traditional analyses, to assess the feasibility of using ML as a method to classify the given GTES force–time curves based on the demographic variables of age (2 classes: old/young) and sex (2 classes: male/female).

### 2.3. Data Analysis Plan

The descriptives were calculated for all variables, including the different parameters of the GTES force–time curve. ANOVAs, independent sample *t*-tests, and correlation analyses were conducted to assess and compare the relationships of the different GTES parameters with GS and demographics such as age, sex, and body mass index (BMI). In the machine learning approach, different supervised ML models (all datasets were labeled for both the target variables) were applied to the GTES force–time curves, and the models’ accuracy and area under the receiver operating characteristic curve (AUC) were calculated and compared. For the ML model analyses, we also used the standard training (single train–test split) as well as cross-validation (3-fold and 5-fold) training approaches.

## 3. Results

### 3.1. Demographics

Thirty-one participants completed the study, and their demographics are presented below in Table 1.

### 3.2. Traditional Analyses

Table 2 below summarizes the different GTES parameters and the GS. The peak GTES, average GTES, and rate of force development (RFD) were lower and the rise time to 80% of the peak was longer among older adults than younger adults. While there was no statistical significance in the difference in any GTES parameters except rise time between older and younger adults, there was a clinically meaningful difference of over 8 N in the peak GTES and over 10 N/s in the RFD. The independent samples *t*-test showed that all the differences in GTES parameters between males and females were statistically significant, with a difference of >13 N in peak GTES, >14 N/s in RFD, and >1.5 s rise time. Males had a higher peak GTES and RFD and a quicker rise time than females. A two-way ANOVA of the GTES parameters showed a lack of significance in the interaction effect of age and sex. However, Table 2 shows that older females have the lowest peak GTES and RFD and the longest rise time, which warrants further investigation.

The correlation analysis results in Table 3 below show that peak GTES correlates more strongly with BMI than GS, especially among older adults (r = 0.594). A significant (*p* < 0.05) moderate positive correlation for the total sample (r = 0.545), older adults (r = 0.519), and younger adults (r = 0.546) was observed in the correlation between peak GTES and GS.

### 3.3. Machine Learning Analyses

The first step in applying ML to characterize the GTES curve was to check whether machine learning classifiers can support the evidence from the traditional analyses by accurately differentiating between age and sex. The accuracy of all three classifiers was the same (66.67%) when age was the target variable. With sex as the target variable, the SVM classifier had the highest accuracy of 66.67%, while the k-NN (five nearest neighbors) and RF classifiers had an accuracy of 55.56% each. By further varying the hyperparameters, such as kernel type, number of nearest neighbors, etc., the k-NN (10 nearest neighbors) classifier had the highest accuracy of 77.78% when sex was the target variable. The accuracy scores and the AUC values for the different models are presented in Table 4 below. The AUC represents the true positive-to-false positive rates in the classification, and values of 0.5 or less indicate a lack of ability of the ML model to distinguish between the classes and is equivalent to random guessing [48].

## 4. Discussion

In this study, we characterized the ToeScale output curve by applying traditional methods based on key features for muscle performance and explored the feasibility of applying ML methods to classify the time series data. To our knowledge, this is the first study to quantify great toe strength beyond just peak force. The key parameters extracted from the GTES force development curve included the peak GTES, average GTES, rise time, and RFD. The results showed statistically significant differences in all GTES parameters between males and females, and the statistically significant differences observed while comparing the GTES parameters by age were in the rise time and RFD. The differences in the peak and average GTES between older and younger adults observed in this study were 7 N and 8 N, respectively, and this is supported by many studies reporting similar trends and differences in great toe strength (GTS) among healthy older and younger adults [28,31]. Existing studies [49,50] also support sex-based differences (~7–8 N), and in this study, males had a higher peak and average GTES than females by 16 N and 10 N, respectively. Another important component of muscle strength that is often overlooked is RFD. Due to its clinical and physiological significance, there has been an increase in the number of studies reporting RFD and its association with muscle function and performance [38,51,52,53,54]. Among the studies reporting the RFD, very few specifically report the RFD of the great toe muscles. Kamasaki et al., 2024 and Sarikaya et al., 2022 [55,56] reported a strong association of RFD in the great toe with better functional mobility and balance outcomes. Additionally, Kamasaki et al., 2024 [55] reported a higher RFD among younger adults compared to older adults, which is consistent with the findings of this study, where the RFD was ~15 N/s higher among younger adults. Although Kamaski et al. did not report any sex-based differences, our study showed that males had 21 N/s higher RFD than females [56]. Finally, our study revealed that older adults had a longer rise time than younger adults. While none of the existing studies report results on rise time, the prolonged rise time in older adults is expected, as muscle responses have been reported to slow down due to aging [57]. This study shows a decline in all GTES parameters with age (Table 1), consistent with studies reporting the effects of age on muscle strength [28,31,55].

The correlation analyses summarized in Table 2 revealed moderate levels of association between the different GTES parameters and BMI, which varied by age, and a low-to-moderate correlation between GS and BMI across all age groups. Among all the GTES parameters, peak GTES had the highest correlation with BMI, particularly among older adults (r = 0.594, *p* = 0.012). The rise time of the RFD was statistically significantly correlated with BMI. The analyses between the different GTES parameters with GS show that all GTES parameters were statistically significantly correlated with GS with moderate levels of association (Table 2). However, when evaluated separately, while the peak GTES was statistically significantly correlated with GS in both younger (r = 0.546, *p* = 0.043) and older adults (r = 0.519, *p* = 0.033), the average GTES (r = 0.516, *p* = 0.034) and RFD (r = 0.473, *p* = 0.055) were more strongly correlated with GS among older adults when compared to younger adults. While BMI and GS are well-established measures commonly used as biomarkers of physical health and mobility and overall muscle strength status [58,59,60], they are known for their low sensitivity to changes in mobility when compared to measures of lower-extremity functions or muscle strength [57]. Studies have reported foot or great toe strength measures to be better predictors of physical health and mobility as they are directly involved in walking [61,62]. The moderate correlations between GTES parameters, especially the peak forces between BMI and GS, highlighted the potential of GTES as a clinical marker for health and functional assessment. The age-based differences in correlations between GTES and BMI and between GTES and GS suggested that changes in BMI and GS are different from changes in GTES with aging.

The three different ML models used to classify the GTES force development curves had moderate accuracies ranging between 55 and 78%, with all models having an accuracy of 66.67% when age was used as the target variable. When sex was used as the target variable, while the SVM had an accuracy of 66.67%, the RF and k-NN (k = 5) models had the lowest accuracy of 55.56%. The model with the highest accuracy was the k-NN classifier with k = 10 with sex as the target variable, which had an accuracy of 77.78%. However, the AUC of this model was 0.36, making this model less reliable for sex-based classification despite the high accuracy. While these models aligned with the results of the traditional analyses, the low AUC scores, particularly for sex-based classification, indicated less effectiveness of the machine learning models in distinguishing between the classes (male/female). The low–moderate AUC values could be attributed to the relatively small sample size and imbalance between males and females [63]. All ML classifiers using age as the target variable had higher AUC scores (0.5–0.75) when compared to ML classifiers using sex as the target variable (0.1–0.5), which further confirmed the impact of the imbalance in sex distribution. Despite the low–moderate AUC scores, these results are promising, and the logical next step in the ML analyses would involve using the manually extracted GTES parameters as “features” to classify by age or sex and compare the accuracy and AUC scores with the results reported in this study.

The current study has several limitations. Firstly, this study has a small sample size, and there is an uneven distribution of males and females included in the study. In addition, we collected only a single trial of GTES and GS measurements, resulting in a smaller dataset for the machine learning methods. The similar values of ML model accuracies and AUC scores across the different models, with and without cross-validation, are indicative of the small and imbalanced dataset analyzed in this study. Secondly, while the results of this study are comparable to the trends in great toe strength reported in existing studies, the extension strength results were difficult to compare with the literature, as most existing studies looking at age- and sex-based differences primarily report on great toe flexion strength.

## 5. Conclusions

The results of this pilot study are promising, as the differences in the GTES parameters are consistent with the current evidence on the age- and sex-based differences in great toe strength and provide more information than just the peak strength and rate of force development. Peak GTES is more strongly correlated with BMI than GS is with BMI. Thus, these findings indicate that different GTES parameters could potentially provide insights, other than GS and BMI, into different physical health- and mobility-related outcomes and how they are affected by aging and other health conditions. Additionally, the ML classifiers with moderate accuracies and low-to-moderate AUC scores are consistent with the results of the traditional analyses. However, future studies with a larger sample size, more methodological rigor, and measurements of both the flexion and extension strength of the great toe with balance and functional mobility measures are warranted to better understand the impact of GTS on mobility and balance.

## Figures and Tables

**Figure 1 sensors-24-04841-f001:**
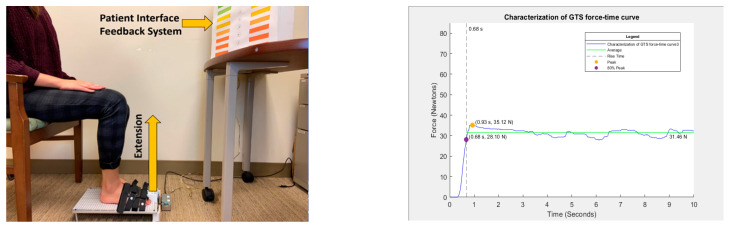
Measurement of GTES using ToeScale (**Left**); parameters selected based on muscle performance to characterize GTES force–time curve (**Right**).

**Table 1 sensors-24-04841-t001:** Demographics of the participants.

Variable	Total Sample	Older Adults	Younger Adults	*p*-Value	Males	Females	*p*-Value
Sample size	31	17	14	NA	9	22	NA
Weight (Kg)	65.29 (13.63)	67.88 (15.05)	62.14 (10.99)	0.250	77.06 (14.38)	60.47 (10.13)	0.002
Height (m)	1.67 (0.10)	1.67 (0.11)	1.66 (0.10)	0.814	1.78 (0.09)	1.62 (0.09)	<0.001
BMI (Kg/m^2^)	23.31 (3.47)	24.04 (3.44)	22.43 (3.42)	0.203	24.15 (3.68)	22.97 (3.41)	0.400

All values of demographics are presented as means (standard deviation).

**Table 2 sensors-24-04841-t002:** Sex- and age-based differences in GTES measures and GS.

All Variables		All	Males	Females	*p*-Value
Peak GTES (N)	Total Sample	42.89 (15.47)	54.82 (13.06)	38.01 (13.82)	0.004
	Older	39.32 (17.25)	58.02 (18.09)	33.57 (12.57)	
	Younger	47.22 (12.22)	52.26 (8.79)	44.42 (13.39)	
Average GTES (N)	Total Sample	34.78 (13.15)	42.45 (10.61)	31.64 (12.98)	0.04
	Older	31.58 (13.64)	44.42 (13.62)	27.63 (11.39)	
	Younger	38.67 (11.84)	40.88 (8.88)	37.44 (13.55)	
Rise Time (s)	Total Sample	2.29 (1.94)	1.02 (0.41)	2.82 (2.09)	0.02
	Older	2.94 (2.22)	1.11 (0.62)	3.5 (2.23)	
	Younger	1.52 (1.21)	0.96 (0.17)	1.83 (1.44)	
Rate of force development (RFD in N/s)	Total Sample	28.64 (21.80)	46.87 (14.76)	21.18 (19.89)	0.002
	Older	21.79 (20.09)	49.17 (19.59)	13.37 (12.21)	
	Younger	36.95 (20.82)	45.02 (11.74)	32.47 (23.91)	
Above Avg GTES (%)	Total Sample	42.59 (13.53)	50.22 (11.59)	39.48 (13.24)	0.04
	Older	36.15 (11.37)	42.45 (10.61)	34.21 (11.27)	
	Younger	50.42 (11.96)	56.44 (8.69)	47.08 (12.63)	
GS_N (N)	Total Sample	259.14 (100.20)	374.85 (66.11)	211.81 (67.77)	<0.001
	Older	239.99 (93.56)	355.37 (89.58)	204.50 (67.76)	
	Younger	282.39 (103.26)	390.44 (45.17)	222.36 (70.39)	

**Table 3 sensors-24-04841-t003:** Correlation between GTES parameters, GS, and BMI.

Variables	Whole Sample ^a,b^		Older Adults ^a^		Younger Adults ^b^	
	GS	BMI	GS	BMI	GS	BMI
Peak GTES	0.545 *	0.390 *	0.519 *	0.594 *	0.546 *	0.297
Average GTES	0.512 *	0.301	0.516 *	0.513 *	0.445	0.219
Rise Time	−0.431 *	0.143	−0.324	0.118	−0.451	−0.103
RFD	0.568 *	0.004	0.473 ^^^	0.152	0.403	0.133
GS	1	0.155	1	0.414	1	−0.012
BMI	0.155	1	0.414	1	−0.012	1

^a^: Lack of normality in RFD; ^b^: lack of normality in rise time; ^a,b^: lack of normality in rise time and RFD; *: statistically significant; ^: trending towards statistical significance.

**Table 4 sensors-24-04841-t004:** Accuracy and AUC scores for the different ML classifiers.

Model	Target Variable	Validation Accuracy (%)	Test Accuracy (%)	AUC
Support vector machine (SVM)	Age	62.5	66.67	0.72
	Sex	75	66.67	0.14
K-nearest neighbors (k-NN, k = 5)	Age	62.5	66.67	0.75
	Sex	87.5	55.56	0.5
K-nearest neighbors (k-NN, k = 10)	Age	62.5	66.67	0.5
	Sex	75	77.78	0.36
Random forest (RF)	Age	62.5	66.67	0.67
	Sex	100	55.56	0.46

## Data Availability

The data presented in this study may be requested from the authors. For deidentified human participant data, contact the corresponding author (H.W.). Institutional approvals and data use agreements may be required. The deidentified data are not yet publicly available because the study is ongoing.

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
