# Peer review of "Characterization of Great Toe Extension Strength Using ToeScale—A Novel Portable Device"

_sensors, 2024, doi:10.3390/s24154841_

Round 1

Reviewer 1 Report

Comments and Suggestions for Authors

Great toe strength was studied in this research using a portable device and machine learning algorithms. 

Major comments:

1. row 73, "resulting in a force-time curve with 500 data points for each participant", if only one 10 second time-series test is done for each subject, That is concerning because what about the discrepancy/inconsistency of multiple trials for one person due to inconsistent motion, fatigue or hysteresis. The researchers must address that how one 10 second time-series sample test data could reasonably represent/characterize a specific subject. 

2. How would sampling frequency (50 Hz) affect the machine learning classifier accuracy? Did you try a different sampling frequency to compare? Since The K-Nearest are basically analyzing different discrete samples, I would expect sampling frequency to make a difference. If that is indeed the case, how would you justify using the 50 Hz sampling frequency? Furthermore, how would you justify the overall results you presented in table.

Author Response

Comments 1: row 73, "resulting in a force-time curve with 500 data points for each participant", if only one 10 second time-series test is done for each subject, That is concerning because what about the discrepancy/inconsistency of multiple trials for one person due to inconsistent motion, fatigue or hysteresis. The researchers must address that how one 10 second time-series sample test data could reasonably represent/characterize a specific subject.

Response 1: Thank you for the feedback. This paper is focused on the preliminary analyses of the pilot data collected with the ToeScale so we did have only a single trial per person, which we have acknowledged as a limitation in the study. We did, however, give all participants 2 minutes to practice and get familiarized with the measurement protocol as well as the device. Additionally, the initial validation study  for the ToeScale (Hile et al, 2023, which is cited in this paper, citation number 35) shows that the device has ICCs of >0.9 for both feet. We are currently working on a larger study with multiple trials for each participant to account for the random errors and within-participant variability. While the single trial of toe strength is not representative of the participant, this pilot data collection helped to determine the feasibility of using the ToeScale in a community setting.

Comments 2: How would sampling frequency (50 Hz) affect the machine learning classifier accuracy? Did you try a different sampling frequency to compare? Since The K-Nearest are basically analyzing different discrete samples, I would expect sampling frequency to make a difference. If that is indeed the case, how would you justify using the 50 Hz sampling frequency? Furthermore, how would you justify the overall results you presented in table.

Response 2: Thank you for the feedback and questions. We did not try an alternate sampling frequency. However, a sampling frequency of 50 Hz was chosen based on the limitations of the hardware as well as based on the Nyquist theorem, and the movement of the great toe during voluntary isometric contraction is estimated to be between 1-5 Hz, which is less than half the sampling frequency. Furthermore, since the sampling frequency is a lot higher than the movement of the great toe, the available data points in each trial (500) would be sufficient for the application of k-NN classifier in this study as the sampling rate is much higher than the actual movement of the great toe thus providing ample data points to prevent loss of information or features while applying the k-NN model. The data presented in the table for the ML analyses are indicative of certain models being better suited for the classification by age/sex over others, which lays the foundation for our future study with multiple trials per participant to chose a more appropriate ML model for the classification. 

Reviewer 2 Report

Comments and Suggestions for Authors

This paper presents a research on using ML classic methods to classify time series obtained using a novel device intended to characterize participants' Great Toe Extension Strength (GTES).

Bellow you can find my comments:

* Line 26, line 29, line 32,… Is ti ok to put the “.” before the bibliography references instead of after them?

* Line 64. “Jamar handgrip dynamometer”. Is it possible to add a reference?

* Line 62. Indicate how many subjects of each age group.

* Line 60. Section 2.1. Does the protocol include repetitions of the measurement for the same participant? 

* Line 60. Section 2.1.The ToeScale portable device is not described. Please describe in more detail.

* Line 60. Section 2.1. Please describe the inclusion criteria with more detail.

* Line 71. Section 2.2. Please include an image showing the diferente features of the GTES curve describe in this section. 

* Line 96. “We applied the k-nearest neighbors (k-NN), support vector machine 96

(SVM), and random forest (RF) as those methods were frequently used models for time- 97

series data.” This argument is weak. Please use a stronger (scientific) argument to support this selection.

* Line 98. “Before running any of the ML models, the GTES data with missing data were 98

imputed based on the time-point within the 10-second trial where the data was missing.” Why do you have missing data? There is no previous explanation about the missing data. Please clarify.

* Line 102. “…ML as a method to classify…” You say you use ML to classify the GTES curves. What is the classification criterion? How many classes do you have? What do those classes mean/represent? Please clarify.

* Section 2. Please explain the training model. Supervised? Unsupervised? Is the dataset annotated/labeled? Please describe in more detail.

* Line 113. I think the description of the data set should be included in section 2.

* Line 116. What are the values into brackets? Please indicate the units. 

* Line 138. For the sake of uniformity between table 2 and 3, I suggest that both tables follow the same arrangement of data (columns for age and rows for data characteristics. i.e. Peak GTES, Average GTES, etc).

* Line 144. It is hard to believe that the accuracy was exactly the same (up to two decimals places) for the three classifiers for the age.

* Section 3.3 How was the validation and test experiment conducted? Did you use cross validation? Please specify.

* Section 3.3. Please describe the AUC.

* Line 165. Please inserta a blank space between “studies” and “[39,40]“.

* Line 171. Do two references comply with the journal’s template way of referencing?

* Line 191. Typo: “(r=519” instead of “(r=0.519”

* Line 194. Typo: “h status.[48–50] They a”. I think the “.” Should be a “,”

* Lines 207 to 212. Do not repeat the section 3.3 explanation.

* Line 229. Typo: “machining” instead of machine.

Round 2

Reviewer 2 Report

Comments and Suggestions for Authors

Dear authors,

The manuscript has been greatly improved after your review.

To my humble opinion, the references are not yet included in the proper way. You put the punctuation symbols before the references instead of before. 

* Line 76-77. “with one column 76 representing time and the second column representing GTES in Kg and the data is saved 77 as a text file, which is then.

Review the phrasing. It is nor correct.

Comments on the Quality of English Language

Please review the wording of paragraph in line 76-77.

Author Response

Comment 1: The manuscript has been greatly improved after your review.

Response 1: Thank you so much for the kind words and for providing feedback in the previous round to help improve our manuscript.

Comment 2: To my humble opinion, the references are not yet included in the proper way. You put the punctuation symbols before the references instead of before.

Response 2: Thank you for the clarification and feedback, the punctuation symbols have now been moved to after the references.

Comment 3: * Line 76-77. “with one column 76 representing time and the second column representing GTES in Kg and the data is saved 77 as a text file, which is then.” Review the phrasing. It is nor correct.

Response 3: Thank you for pointing this out, it has been corrected.